# Hydrogels Based on Polyacrylamide and Calcium Alginate: Thermodynamic Compatibility of Interpenetrating Networks, Mechanical, and Electrical Properties

**DOI:** 10.3390/biomimetics8030279

**Published:** 2023-06-28

**Authors:** Alexander P. Safronov, Nadezhda M. Kurilova, Lidiya V. Adamova, Tatyana F. Shklyar, Felix A. Blyakhman, Andrey Yu. Zubarev

**Affiliations:** 1Institute of Natural Sciences and Mathematics, Ural Federal University, 620002 Ekaterinburg, Russia; nadyakurilova98@yandex.ru (N.M.K.); lidiya.adamova@urfu.ru (L.V.A.); t.f.shkliar@urfu.ru (T.F.S.); feliks.blyakhman@urfu.ru (F.A.B.); a.j.zubarev@urfu.ru (A.Y.Z.); 2Institute of Electrophysics UB RAS, 620016 Ekaterinburg, Russia; 3Department of Biomedical Physics and Engineering, Ural State Medical University, 620028 Ekaterinburg, Russia

**Keywords:** interpenetrating networks, alginate, polyacrylamide, swelling ratio, thermodynamic compatibility, dynamic mechanical analysis, Donnan potential

## Abstract

The synthesis and physicochemical properties of hydrogels with interpenetrated physical and chemical networks were considered in relation to their prospective application as biomimetic materials in biomedicine and bioengineering. The study was focused on combined hydrogels based on natural polysaccharide—calcium alginate (CaAlg) and a synthetic polymer–polyacrylamide (PAAm). The series of hydrogels with varying proportions among alginate and polyacrylamide have been synthesized, and their water uptake has been characterized depending on their composition. The equilibrium swelling and re-swelling in water after drying were considered. The compatibility of alginate and polyacrylamide in the combined blend was studied by the thermodynamic approach. It showed a controversial combination of negative enthalpy of mixing among PAAm and CaAlg with positive Gibbs energy of mixing. Mechanical and electrical properties of the combined gels with double networking were studied as relevant for their prospective use as scaffolds for tissue regeneration and working bodies in actuators. The storage modulus and the loss modulus were determined in the oscillatory compression mode as a function of proportions among natural and synthetic polymers. Both moduli substantially increased with the content of CaAlg and PAAm. The electrical (Donnan) potential of hydrogels was measured using the capillary electrode technique. The Donnan potential was negative at all compositions of hydrogels, and its absolute values increased with the content of CaAlg and PAAm.

## 1. Introduction

Polymeric hydrogels are a type of biomimetic material that perfectly meets the demands and requirements of the development of advanced biotechnology and bioengineering [1,2,3,4]. Hydrogel is, essentially, a binary system that contains a cross-linked polymeric network with one component swollen in liquid water being the other component. Internally, a hydrogel is a polymer solution; the linear sub-chains of the polymeric network take equilibrium conformations as those in liquid solutions of linear polymers. However, due to the integrity of a cross-linked network, the gel is not fluid but a soft elastic material with definite shape and boundaries. In essence, the properties of the hydrogel are governed, on the one hand, by the structural arrangement of the polymeric network, and on the other hand, by the molecular interactions among the components of a gel.

There are two major routes to making a polymeric network in a liquid medium. One is referred to as the “physical” cross-linking of macromolecules. This way is typical for biopolymers: polysaccharides and structural proteins like gelatin or actin. In physical gelation, long polymeric macromolecules are linked together due to the aggregation of monomeric units of neighboring chains. These aggregates usually have a quasi-crystalline helical structure stabilized by inter-chain H-bonds [5,6,7]. Physical hydrogels are biocompatible, but their internal network structure is typically inhomogeneous, and their properties are, to a certain extent, irreproducible due to the structural variability of biopolymers. The gelation of physical gels occurs stochastically, and it is difficult to maintain their networking density. Therefore, the mechanical properties of physical gels can be controlled in a very limited way.

The other route to gel formation is typical for synthetic polymers. Networking takes place in the synthesis of polymer from monomers in the presence of special chemical agents which provide chemical cross-linking simultaneously with the growth of linear chains. The networking density can easily be controlled by maintaining a definite ratio among the monomers, which constitute the linear chains and the cross-linker. The cross-links in the network are chemical bonds that provide stability to the gel structure. The structure and performance of chemical gels are well reproducible and can be controlled by changing composition at the synthesis. Meanwhile, chemical gels are less biocompatible and less biomimetic than physical gels.

The advantages of biocompatible physical networks and synthetic chemical networks can be combined in hydrogels with interpenetrating networks (IPN) [8,9,10]

Polyacrylamide (PAAm) is often used as a polymer to constitute the chemical network as it is easily polymerized in a wide range of monomer concentrations and networking densities. Polyacrylamide is compatible with many biopolymers, which can form physical networks [11]. The properties and applications of IPNs based on acrylamide are presented in the literature [7,12,13,14].

Alginate is one of the polysaccharides that can successfully be combined with PAAm in IPN gels with a double chemical/physical network. Alginic acid hydrogels are widely used in the conditioning of fabrics, foods, and various drug delivery systems [15,16,17,18,19,20,21,22]. They can be easily obtained by the K^+^/Ca^2+^ ionic exchange. The shortcomings of alginate gels are fast biodegradation and poor mechanical performance. The combination of alginates with chemically cross-linked polyacrylamide enhances their stability decreases structural heterogeneity, and improves the mechanical performance of hydrogels [23,24,25,26]. IPN hydrogels with a chemical network of PAAm and a physical network of alginate are tested in various biomedical applications [10,27,28,29,30,31,32,33].

The objective of the present study was to focus on the basic physicochemical features that form the background of the application of IPN hydrogels with the PAAm chemical network and alginate physical network. IPN gels with varying concentrations of these two polymeric components were considered, and characterization of the network structure and the thermodynamic compatibility among PAAm and CaAlg interpenetrating networks was provided. Mechanical and electrical properties of PAAm/CaAlg hydrogels are presented with respect to the composition of the double network of these hydrogels.

## 2. Materials and Methods

### 2.1. Synthesis of PAAm/CaAlg Hydrogels

Hydrogels with interpenetrating chemical networks of polyacrylamide (PAAm) and calcium alginate (CaAlg) were synthesized using a two-step route. First, polymerization of the chemical network of PAAm was done via radical polymerization in a water solution. Mixed solutions in water were prepared, which contained monomer acrylamide, purity ≥99% (AAm) (AppliChem, Darmstadt, Germany), cross-linker—N,N′-methylene-bis-acrylamide, purity ≥99.5% (MBAAm) (Merck, Schuchardt, Honenbrunn, Germany), sodium alginate (NaAlg) (Sigma Aldrich, St. Louis, MI, USA), initiator—ammonium persulfate, purity ≥98% (APS), (Merck, Schuchardt, Honenbrunn, Germany). Concentrations of AAm in solution were set to 0.8 M, 1.6 M, and 3.2 M. Concentrations of NaAlg were set to 0, 1, 3, and 5% (wt.). Thus, a series of 12 compositions were prepared. The concentration of MBAAm was always kept at a 1:100 molar ratio to AAm. Solutions were vigorously stirred until complete dissolution of NaAlg occurred. Then APS was added to solutions in a 3 mM concentration. Reaction mixtures were placed in cylindrical polyethylene tubes with an 8 mm diameter. Polymerization took place in the thermostat at 80 °C for 1 h. After that, cylindrical gels were removed from molds and placed in CaCl_2_ 0.5 M water solution for 3 days to form a physical network of CaAlg due to ionic exchange with Na^+^ cations. The resulting gels with a double network (DN) were then washed in an excess of distilled water for 2 weeks with daily water renewal to remove the salts and traces of reactants and to achieve the equilibrium swelling of hydrogels. The synthesized gel will further be denoted as PAAmX/CaAlgY, whereas “X” stands for the molar concentration of AAm, and “Y” is the weight % of NaAlg in the reaction mixture. For instance, PAAm0.8/CaAlg1 denotes hydrogel synthesized in 0.8 M solution of AAm with 1% of NaAlg added. Individual CaAlg hydrogel for re-swelling measurements and for thermodynamic studies was prepared separately. NaAlg water solution (3%) was poured onto the Petri dish in a 3 mm layer, and 0.5 M CaCl_2_ solution was added on top of it. The obtained gel plate ca. 3 mm thick was kept in 0.5 M CaCl_2_ solution for 3 days and then washed in distilled water for a week with daily water renewal to remove salts. Then the plate was cut into 5 × 5 mm pieces for reswelling measurements. For thermodynamic measurements, CaAlg and PAAm/CaAlg hydrogels were dried in a thermostat at 80 °C to the constant weight.

### 2.2. Methods

The equilibrium water uptake (swelling ratio) of hydrogels was determined gravimetrically. The swollen gels were weighed; then they were dried to constant weight at 80 °C, and the dry residue was weighed again. The swelling ratio was calculated according to the Equation:(1)$$\alpha =\frac{m-m_{0}}{m_{0}},$$

Here, *α* is the equilibrium swelling ratio, *m* is the mass of the swollen gel, and *m*_0_ is the mass of the dried gel.

The enthalpies of swelling of hydrogels in water were measured using a 3D Calvet calorimeter SETARAM C80 (SETARAM, Caluire, Auvergne, France) at 25 °C. A pre-weighted amount of dry gel was placed in a thin glass ampoule, which was then sealed with a burner. Seven milliliters of water were placed into a stainless-steel calorimetric cell. The sealed ampoule was placed there in a special holder. The assembled cell was mounted in a calorimeter and thermally equilibrated overnight. The experiment started by breaking the ampoule in water inside the cell using a breaking rod of an ampoule vessel. The heat evolution curve was then recorded. The typical experiment lasted 2 h until the initial baseline was re-established. Integration of the heat evolution curve was performed using the integrated CALISTO software. The error of the measurement of enthalpy of swelling was less than ±0.02 J.

The sorption of water vapor by hydrogels at 25 °C was studied using the volumetric method using a Micromeritics ASAP 2020 automatic surface area and porosity analyzer with an attachment for vapor sorption (Micromeritics, Norcross, GA, USA). The pre-weighted amount of dry gel was placed in a glass ampoule 5 mL in volume and additionally vacuum-dried for 3 h at 70 °C. Then, the ampoule was connected to a vacuum port of the analyzer, and the volume of the ampoule was calibrated. Water vapors were dosed stepwise to the ampoule under the permanent control of vapor pressure in the ampoule. At each step, a certain vapor pressure was first applied to the ampoule, and then the pressure gradually decreased to an equilibrium level due to the sorption of vapor by the gel sample. The amount of adsorbed water was proportional to the decrease of vapor pressure in the ampoule at each step of the experiment.

A laboratory setup for mechanical tests [34] was used to evaluate the viscoelasticity of hydrogels. Cylindrical samples were placed between two plates. One was connected rigidly to an actuator of a linear electromagnetic motor, and the other was connected to a precision force transducer. The motor-induced compression strain had a sinewave mode with a magnitude of up to 15% of the initial length of the gel cylinder. The elastic storage modulus (*E*′), the loss modulus (*E*″), and the loss tangent (*tgδ*) were calculated at a frequency range of 0.01–20.0 Hz.

The testing of the electrical properties of gels was performed using a laboratory installation for the electrical potential measurement in biophysical systems described elsewhere [35]. The measurement of the electrical potential of hydrogel was performed with two identical Ag/AgCl tapered glass microelectrodes (~1 micron in tip diameter) typically used in biophysical studies for intracellular voltage measurement. The electrodes were single-pulled using a standard electrode puller, ME-3 (EMIB Ltd., Moscow, RF), from thin-walled, single-barrel borosilicate capillary tubes, TW150F-6 (World Precision Instruments, Sarasota, FL, USA). The pulled electrodes were immersed in a 3 M KCl solution with the tip facing upward so that the solution filled the tip through capillary action. One electrode was pinned into the ferrogel sample, and the other was placed into outside water. The potential difference between microelectrodes was measured using an instrumental amplifier on the base of an integrated circuit, INA 129 (Burr-Brown, Dallas, TX, USA). To reduce the influence of electromagnetic interference on the potential difference measurement, special wire shields were provided around the measuring unit.

## 3. Results and Discussion

### 3.1. Swelling of Double-Network Hydrogels

The equilibrium swelling ratio of hydrogels, or in other words, their equilibrium water uptake, provides the fundamental characterization of their inner networking structure. The swelling ratio is a function of both the networking density of a gel and the balance between water/water, water/polymer, and polymer/polymer molecular interactions in the interior. Figure 1 presents the dependence of the swelling ratio of PAAm/CaAlg DN hydrogels on the concentration of the AAm monomer and the Alg solution in the reaction mixture for the polymerization of DN hydrogels. The numerical values are given in Table 1.

The general trend in Figure 1 is the diminishing of the swelling ratio with the increase in the concentration of alginate and PAAm both, which is the typical type of behavior for hydrogels. In the case of the physical network of CaAlg, the growth in polymer concentration favors the formation of inter-chain aggregates, which are the source of physical cross-links. Concerning CaAlg gels, these are the ionic complexes among Ca^2+^ cation and four α-L-guluronic (G) residues in the alginate chain. Two adjacent G residues belong to one interacting chain; the other pair of adjacent G residues belong to a neighboring chain. As a result, these multiple chelate complexes form the “egg-box” structure [15,16,17] of CaAlg physical networking.

In the case of the PAAm chemical network, the influence of monomer concentration is indirect. Initially, the monomer-to-cross-linker ratio for all gels was set to 100:1, which presumably provided the same networking density. Polymer entanglements are more likely to happen at a higher AAm monomer concentration, increasing the actual crosslinking density by introducing spatial cross-links in addition to chemical cross-links.

The common approach to evaluating the density of the chemical networking is based on the Flory–Rehner equation [36], which relates the average number of monomeric units (*N_C_*) in linear subchains between adjacent cross-links of the network to the volume fraction of polymer in a swollen gel. In terms of the volume swelling ratio, it gives [37]:(2)NC=V1α02αp−α0αp13V2ln⁡1−αp−1+αp−1+χαp−2
where *V*_1_ and *V*_2_ are the molar volumes of the solvent and polymer, respectively, *χ* is the Flory–Huggins parameter for the polymer-solvent system, *α*_0_ is the swelling ratio of gel as provided by the composition of the reaction mixture in the synthesis, *α_P_* is the actual swelling ratio of gel.

Equation (2) was used to estimate the density of the PAAm chemical network both in individual PAAm hydrogels with chemical networking and in combined DN hydrogels. In the latter case, however, Equation (2) cannot be used directly as the swelling ratio of DN hydrogels as it relates to the combination of the networks: chemical (PAAm) and physical (CaAlg). To elaborate on Equation (2) in this case, we have considered DN hydrogel in a simplified way as a filled polymer composite. The PAAm flexible chemical network comprised the polymeric matrix of this composite filled with rigid polymeric filaments of CaAlg. This is equivalent to the assumption that the conformational mobility of PAAm chains in DN hydrogel dominates over the mobility of semi-rigid Alg chains, and the latter can be neglected. Thus, the whole water uptake of DN hydrogel can be solely related to the swelling of the PAAm chemical network. Of course, such a rough approximation can only describe relative trends in chemical networking for PAAm/CaAlg DN hydrogels.

The volume swelling ratio in relation to the PAAm network can be calculated using the Equation:(3)$$\alpha^{\prime }=\frac{\alpha \rho }{\gamma }$$

Here, *γ* is the weight fraction of PAAm in the polymeric mixture PAAm + CaAlg, which forms the double network of gels; *r* is the density of PAAm.

We have used *V*_1_ = 18 cm^3^/mol (water), *V*_2_ = 56.2 cm^3^/mol (PAAm). The density of PAAm (1.27 g/cm^3^) was calculated as the ratio of molecular weight and molar volume of the AAm monomer unit. The Flory–Huggins parameter *χ* = 0.12 for the water solution of PAAm was calculated in a conventional way [38] based on the solubility parameters of water (24.9) and PAAm (29.0). The same value of the Flory–Huggins parameter was used in the case of combined hydrogels. It is worth noting that rigorous consideration of binary interactions in tri-component systems should include three binary parameters: polymer1/solvent, polymer2/solvent, and polymer1/polymer2. Therefore, the use of the same binary Flory–Huggins parameter both for PAAm and combined DN hydrogels is also the simplification of the “composite-like” model of their structure. Considering all these assumptions, *N_C_* values for combined hydrogels should be considered not accurate but effective.

Table 1 presents the apparent *N_C_* values of the chemical network in DN hydrogels with different PAAm and CaAlg concentrations.

At the highest concentration of PAAm (3.2 M in the synthesis), the parameter of chemical networking (*N_C_*) is almost insensitive to the presence of the physical network of CaAlg. The influence of the CaAlg network became noticeable if the concentration of PAAm had been decreased in the synthesis down to 1.6 M. In that case, the addition of the CaAlg network resulted in an almost two-fold decrease in *N_C_* value. The lowest concentration of PAAm in the synthesis (0.8 M) showed the drastic declination of *N_C_* value upon the formation of the physical network of CaAlg. Most likely, the impact of CaAlg on the network density of PAAm is the result of extra cross-linking provided by inter-chain molecular interactions between PAAm and alginate chains, which will be discussed below. The effect of inter-chain interaction between these two polymers in a combined gel structure is more distinct in a loose network, which provides opportunities for changes in their macromolecular conformations. The dense network implied restrictions on such changes, and the effect of additional physical networking vanished.

The important feature of polymeric hydrogels is their ability to recover water uptake after drying. In this respect, physical and chemical networks showed the opposite behavior. The individual chemical network of PAAm gel almost completely restored its swelling after being dried. The extent of the reverse swelling was 77%, 91%, and 99% for PAAm0.8, PAAm1.6, and PAAm3.2, respectively.

On the contrary, the swelling of individual CaAlg gel was almost irreversible. For instance, the initial equilibrium swelling ratio of CaAlg gel prepared from 3% NaAlg solution was 20.5. It dropped to 5.4 if the gel was completely dried and then placed in an excess of water. The degree of the reverse swelling of CaAlg gels was, however, the function of the percentage of water eliminated from the gel during its drying. Figure 2 presents the dependence of the extent of the reverse swelling (circles) on the percentage of water left in the gel during partial drying. The plot is concave upwards, which means that the partly dried CaAlg gels absorbed water being placed in an excess of it. Another plot (triangles) provides the gain in swelling if the partly dried CaAlg gel was placed in water.

These data indicate the additional cross-linking in the networking structure of CaAlg during the elimination of water, which prevented the swelling recovery. Most likely, these cross-links are the same ionic complexes as Ca^2+^ cations which were favored by the aggregation of alginate chains while drying.

Hence, to provide a recovery of swelling for CaAlg physical gels, the close contact between alginate chains is to be prevented. Presumably, the interpenetrating chemical network might do it. This is confirmed in Figure 3, which gives the dependence of the swelling recovery on the composition of combined PAAm/CaAlg gels. It is obvious that combined gels show very high swelling recovery. It was certainly the result of the restrictions implied by the separate polymeric networks upon each other. Each of them prevented the close contact of the macromolecular chains of the counterpart. Thus, the additional ionic complexes with Ca^2+^ cations did not appear, and the initial networking structure of CaAlg was effectively preserved.

### 3.2. Thermodynamic Compatibility of PAAm and CaAlg in Binary Blend

The evaluation of the intensity of polymer/polymer molecular interactions in the combined network of PAAm/CaAlg hydrogel is not straightforward. In the tri-component system, PAAm/CaAlg/water, at least six separate types of interactions are present: PAAm/PAAm, CaAlg/CaAlg, H_2_O/H_2_O, PAAm/CaAlg, PAAm/H_2_O, CaAlg/H_2_O. In addition, it is apart from the diversity of H-bonding in bulk water and cooperative hydrophobic interactions as well. In such a complex system, some approximations are to be applied.

In general, two main approaches to characterize thermodynamic compatibility in polymeric systems might be distinguished. One is the study of their structure by means of light or neutron scattering with further analysis using theoretical models, which include thermodynamic parameters of interaction among the components. A good example of such an approach applied to DN hydrogels might be found in ref [39].

The other is the direct measurement of thermodynamic functions of mixing for polymeric systems. Examples of such an approach applied to binary blends of synthetic and natural polymers might be found in refs [40,41]. In the present study, we have used it to estimate PAAm/CaAlg interchain interaction in their binary blend.

A swollen DN PAAm/CaAlg hydrogel, due to the variety of different pair interactions, is too complex for direct thermodynamic measurements, which are more applicable to binary systems. Therefore, we have focused exclusively on the pair interactions between PAAm and CaAlg polymeric chains and have restricted consideration to the binary PAAm/CaAlg blend, excluding H_2_O. In such a binary system, interaction among polymers can be characterized by the thermodynamic functions of their mixing defined by the Equation:PAAm + CaAlg = PAAm/CaAlg + (∆*H_m_*, ∆*G_m_*, *T*∆*S_m_*),(4)
these functions (∆*H_m_*, ∆*G_m_*, *T*∆*S_m_*) are, by definition, the thermodynamic gain or the thermodynamic loss if the contacts between similar chains (PAAm/PAAm and CaAlg/Ca/Alg) are replaced by the contacts between different chains (PAAm/CaAlg).

Equation (4) cannot be used for the determination of thermodynamic quantities directly, as individual polymers do not mix with each other. In this case, the common approach [40,41] is the use of the appropriate thermodynamic cycle, which in the case of the PAAm/CaAlg mixture is as follows.

Step (5) in this cycle gives the desired value of the thermodynamic function of mixing in a binary polymer blend according to the Equation:Step (5) = step (1) + step (2) + step (4) − step (3),(5)

To use this Equation, the corresponding thermodynamic functions for steps (1)–(4) are to be evaluated in the appropriate thermodynamic experiment. Steps (1), (2), and (3) (Figure 4) can be accomplished by the swelling of dried gels: PAAm, CaAlg, and PAAm/CaAlg in water. Steps (4) and (5) certainly cannot be performed as the cross-linked networks cannot mix either in their dry or swollen state. Therefore, steps (4) and (5) are given by dashed lines in Figure 4. Step (5) makes no problem for consideration as it provides the calculated value for the thermodynamic function of mixing (∆*H_m_*, ∆*G_m_*, *T*∆*S_m_*), and it needs not be done in the actual experiment. Step (4) is different as it provides the necessary thermodynamic value for calculation using Equation (5). It is worth noting that such a problem does not exist for blends of linear polymers. In this case, step (4) in the thermodynamic cycle (Figure 4) is just the direct mixing of two polymer solutions. As it was shown in refs [40,41] for the mixing of solutions of linear synthetic polymer and linear polysaccharide, the thermodynamic values at step (4) were within experimental errors of thermodynamic values at steps (1), (2), and (3). Here, we supposed the same for the virtual mixing of swollen PAAm and CaAlg networks at step (4) in the thermodynamic cycle, assuming that hydrogel is thermodynamically similar to a dilute polymer solution. According to this assumption, step (4) in Equation (5) was neglected and was not considered in thermodynamic analysis. However, the similarity between solutions of linear polymers and polymeric gels is not universal, and it might, in principle, affect their thermodynamics [39]. Therefore, all thermodynamic values presented below contain such uncertainty.

Thermodynamic functions (∆*H_m_*, ∆*G_m_*, *T*∆*S_m_*) calculated with the use of the cycle given in Figure 4 and Equation (5) refer to the temperature at which thermodynamic experiments at steps (1), (2), and (3) are performed. In the present study, it was performed at 25 °C.

In the case of the enthalpy of interaction (∆*H_m_*) between PAAm and CaAlg, steps (1), (2), and (3) corresponded to the enthalpy of swelling of dry PAAm, CaAlg, and PAAm/CaAlg gels in an excess of water. These values were measured in direct calorimetric experiments. The study was restricted to the gels with 1% alginate and different concentrations of AAm in the reaction mixture. Measured values for the enthalpy of swelling are given in Table 2. There is also given the value of the enthalpy of mixing (∆*H_m_*) among PAAm and CaAlg calculated per 1 mol of PAAm monomer units.

The enthalpy of interaction between PAAm and CaAlg in their binary blend was found to be negative, and within the limits of accuracy, it was the same for all compositions under study. In general, the negative sign of the enthalpy of interaction indicates that interaction between these two polymers is energetically favorable. Both polymers have polar residues in the composition of their monomer units which can provide the formation of inter-chain H-bonds. PAAm monomer unit contains amide residues with C=O and NH_2_ groups; the CaAlg monomer unit contains two O–H residues.

Considering the absolute value of ∆*H_m,_* one should keep in mind that it is an excess thermodynamic quantity, or in other words, it is a gain over the energy of cohesion of each of these two polymers. This means that the inter-chain contacts between PAAm and CaAlg are more energetically favorable than the contacts between similar monomer units in individual polymers. At the same time, the absolute value of ∆*H_m_* is substantially lower than the thermal energy (*RT*).

Based on these results, considering the interaction between PAAm and CaAlg subchains in the combined chemical/physical network of hydrogels, we may assume that the aggregative structures among PAAm and CaAlg subchains are energetically favorable, but they are not stable and can be disrupted by thermal motion.

The Gibbs energy of interaction between PAAm and CaAlg was determined using the same thermodynamic cycle (Figure 4, Equation (5)). In this case, steps (1), (2), and (3) in Equation (5) relate to the change of the chemical potential *(*∆*μ_2_*) of the polymer network during its swelling. Its value cannot be measured directly. Evaluation of ∆*μ_2_* can be completed based on the water sorption isotherms, which were determined for dried CaAlg, PAAm, and PAAm/CaAlg gels. Figure 5 presents an example of such isotherms for CaAlg and PAAm3.2/CaAlg gels. The abscissa in these plots is the relative pressure of water vapor; the ordinate is the amount of adsorbed water in g per 1 g of dry polymer at 25 °C.

Isotherms in Figure 5 have a typical concave shape: water adsorption increases with the relative pressure of water vapor. The highest adsorption was observed for CaAlg dried gel; the lowest was for PAAm3.2 gel. Sorption of water vapor on combined PAAm/CaAlg gel had intermediate values. Similar trends were observed for gels with 1.6 M and 0.8 M concentrations of AAm in synthesis.

The isotherms were converted to the concentration dependencies of the chemical potential of water *(*∆*μ*_1_) in corresponding hydrogels. Therefore, the values of relative pressure (*P/P_S_*) were transformed to the change of chemical potential of water *(*∆*μ*_1_) according to the Equation:(6)$$\Delta \mu_{1}\left(a\right)=\frac{RT}{M_{1}}ln\left[\frac{P}{P_{S}}\right]$$

Here, *M*_1_ is the molar mass of water; *a* is the water adsorption. Equation (6) gives the dependence of the chemical potential of water (per 1 g) on the amount of water in a partly swollen gel. This dependence was used for the calculation of the chemical potential of polymer *(*∆*μ*_2_) according to the Gibbs–Duhem equation, which in its integral form reads:(7)$$\Delta \mu_{2}\left(a\right)=\int_{-\infty }^{\Delta \mu_{1}}a\left(\Delta \mu_{1}\right)d(\Delta \mu_{1})$$

Integration was performed using the spline approximation of *a(*∆*μ*_1_) dependencies. Thus, the concentration dependencies of the specific chemical potential of polymers (per 1 g of polymer) were obtained.

To elaborate on the thermodynamic cycle in Figure 5, we used the values of ∆*μ*_2,_ which corresponded to the maximal adsorption of water. These values are given in Table 2. Using them, the values of the Gibbs energy of mixing (∆*G_m_*) among PAAm and CaAlg were calculated according to Equation (5) at different ratios of CaAlg and PAAm in binary composition. The values of ∆*G_m_* are provided in Table 2 in relation to 1 mol of PAAm monomer units.

As can be seen in Table 2, the chemical potentials of polymers are negative in all cases, but the resulting values of Gibbs energy are positive. It means that the combination of two networks—the chemical network of PAAm and the physical network of CaAlg in a binary polymer blend—is thermodynamically unfavorable. At the same time, as it was shown above, the enthalpy of interaction was negative (Table 2). Hence, it is reasonable to conclude that the thermodynamic incompatibility between PAAm and CaAlg stemmed from entropic contribution. The entropy of mixing (*T*∆*S_m_*) was calculated using a general thermodynamic equation:∆*G_m_* = ∆*H_m_* − *T*∆*S_m_*,(8)

The values of *T*∆*S_m_* are given in Table 2; they were found to be negative. Negative entropy of mixing, in general, indicates restrictions that imply the stochastic distribution of components. Thus, we may assume that the likely reason for the thermodynamic incompatibility of PAAm and CaAlg in their binary blend is the conformational inconsistency of these polymers. PAAm is a typical acrylic polymer with a flexible chain. The Kuhn length for acrylates typically is 1.5–2.5 nm [42], depending on the composition of the monomer. Alginate is a polysaccharide with a semi-flexible chain and a persistent length of 12 nm [43]. Taking the Kuhn length as persistent double length, one obtains 24 nm for alginate, which is an order of magnitude larger than that of PAAm. Such large differences in flexibility lead to entropic losses in the binary mixture, which apparently prevail over the enthalpy of interaction.

Some dependence of ∆*G_m_* on the composition of the PAAm/CaAlg blend may be marked out. The positive values of ∆*G_m_* are the same for PAAm0.8/CaAlg1 and PAAm1.6/CaAlg1 compositions within the accuracy of their evaluation. The values for PAAm3.2/CaAlg1 are larger. Considering conformational restrictions as a major source for the positive values of ∆*G_m,_* it looks reasonable as the content of PAAm in the composition increases.

It is an open question whether these thermodynamic features of the binary mixture of PAAn and CaAlg networks preserve in the case of their interpenetrating networks swollen in water. It might not be necessarily so in the presence of a third component—water—which strongly interacts with both polymers. Meanwhile, we may suppose that these features might be related to the local structuring of interpenetrating PAAm and CaAlg subchains in their close vicinity. In general, one should expect certain energetic gain at PAAm/CaAlg contact but substantial entropic losses as well.

### 3.3. Mechanical Properties of Hydrogels

Mechanical properties of PAAm/CaAlg hydrogels were characterized by means of dynamic mechanical analysis (DMA) in a uniaxial contraction mode at 25 °C. Figure 6 presents the typical frequency dependence of storage modulus (*E*′), loss modulus (*E*″), and loss tangent (*tgδ*) for PAAm1.6/CaAlg0 and PAAm1.6/CaAlg5 hydrogels. Similar dependencies were obtained for other hydrogels. The frequency range of 0.01–20 Hz covers the ambient conditions for mechanical activity in living systems [44].

In this frequency range, almost constant values of the storage modulus were observed (Figure 6a) both for the hydrogel with a chemical network (PAAm1.6/CaAlg0) and the hydrogel with a double network (PAAm1.6/CaAlg5). At the same time, *E*′ values for the hydrogel with a double network were an order of magnitude larger. The loss modulus was also substantially larger for the hydrogel with a double network. The moderate enlargement of *E*″ at lower frequencies was observed for PAAm1.6/CaAlg0 hydrogel. In the case of the PAAm1.6/CaAlg5 hydrogel, the loss modulus remained constant. At higher frequencies, the values of *E*″ substantially decreased for both hydrogels. At any frequency, *E*′ remained substantially larger than *E*″ and there was no tendency to cross-over in this frequency range. It indicated that the deformation of hydrogels was predominantly elastic.

The elastic type of deformation led to low values of loss tangent (*tgδ*) in all frequency ranges (Figure 6b). It is noticeable that *tgδ* dependence on frequency for PAAm1.6/CaAlg0 hydrogel with chemical networking had achieved a maximum at approximately 5 Hz. Such a maximum at-frequency dependence typically corresponds to resonance with some sort of internal movement in the polymer network structure. The characteristic time (the reciprocal of the frequency) for this sort of motion might be estimated by 0.2 s. This maximum was not observed at *tgδ* dependence for PAAm1.6/CaAlg5 DN hydrogel. Apparently, the incorporation of a rigid physical network of CaAlg had blocked the fast motion of a more flexible chemical network of PAAm.

The dependence of hydrogel composition on the dynamic moduli was studied at a frequency of 0.05 Hz. Figure 7 provides a plot of the dependence of the storage modulus on the concentration of PAAm and alginate in the combined hydrogel.

Figure 7 shows a distinct enlargement trend in *E*′ values with both PAAm and CaAlg concentration. In general, it is consistent with an increasing classical trend in the dependence of modulus on the polymer volume fraction in a swollen gel [38]. However, in the case of combined PAAm/CaAlg gels with a double network, no universal dependence of *E*′ on total polymer concentration was observed. Experimental points did not match any curve but were rather dispersed stochastically.

It is noticeable in Figure 7 that the influence of CaAlg concentration on the storage modulus is much stronger than the influence of PAAm concentration. The reason for the dominating impact of CaAlg concentration might be explained based on the thermodynamic features of PAAm/CaAlg compositions which were presented in the previous section. As shown above, the adhesion of PAAm monomer units onto CaAlg chains was energetically favorable. Therefore, we may expect the formation of additional adhesive cross-links in the double network. At the same time, it was shown that the combination of PAAm and CaAlg was thermodynamically unfavorable due to large entropic losses. Most likely, it led to extra stress applied to the conformation of rigid CaAlg subchains in the combined network. Both effects—the inter-chain adhesion and the conformational restrictions—favored the increase in the compression modulus. Such a combined effect might resemble the effect of fiber-loading into polymer composite on its mechanical properties.

The loss modulus (*E*″) of combined PAAm/CaAlg hydrogels increased with both PAAm and CaAlg concentration as well (Figure 8). The impact of CaAlg concentration on *E*″ was also more pronounced than the influence of PAAm concentration. A comparison of the data in Figure 7 and Figure 8 shows that the values of *E*″ at any hydrogel composition remained substantially lower than the values of *E*′. Thus, in any composition of combined hydrogels with a double network, the elastic deformation dominates over the viscous one.

### 3.4. Electrical Potential of PAAm/CaAlg Hydrogels

Hydrogels made of synthetic polymers, in many cases, possess definite electrical potentials, such as living cells and biological tissues [45]. The potential originates from the Donnan equilibrium [46] at the boundary of hydrogel immersed in water. Donnan equilibrium at the boundary is established if at least one sort of ion present in the system cannot move across this boundary. In the case of anionic polymeric gels, these ions are negatively charged acidic residues attached to the macromolecular network. As a result of such restriction in the mobility of macro-ions of the network, the concentration of mobile ions also becomes different between the interior of the gel and the surrounding solution. In the case of anionic gels, the concentration of mobile cations (counterions) is larger in the interior of the gel than in the supernatant. This difference provides a negative value of Nernst potential across the boundary of a gel. Conventionally, this drop in electrical potential is defined as Donnan potential.

Figure 9 presents the dependence of Donnan potential of combined PAAm/CaAlg hydrogels with a double network in their composition.

All studied hydrogels showed negative values of Donnan potential. The absolute values enlarged with both the concentration of PAAm and the concentration of CaAlg.

Alginic acid is an anionic polyelectrolyte, as well as its salts. It contains carboxylate groups in each monomeric unit. The polyelectrolyte nature of alginates is more pronounced for soluble salts like NaAlg. By contrast, CaAlg is insoluble in water, but nevertheless, limited dissociation of it is to be expected. Thus, some carboxylate residues in CaAlg gel are not bound in complexes with Ca^2+^ ions but exist in the form of anions. These ionized carboxylate residues provide a negative net charge of polymeric subchains in the physical CaAlg gel and give rise to its negative Donnan potential. Thus, with the growth of CaAlg concentration in combined PAAm/CaAlg DN hydrogels, the absolute value of Donnan potential should also increase as the concentration of ionized carboxylate residues enlarged.

The influence of PAAm concentration on the Donnan potential of combined PAAm/CaAlg gels cannot readily be explained. In general, PAAm is a neutral polymer, and presumably, it should not provide Donnan equilibrium and Donnan potential on its boundary. However, PAAm gels possessed negative Donnan potential, which increased in absolute values with PAAm concentration in the network. The same trend occurred in combined PAAm/CaAlg gels. In this respect, several options might be considered to explain this effect. The first option is a tendency of amide groups in PAAm monomer units to hydrolysis, which should result in anionic carboxylate residues. Their concentration can be small but still sufficient for negative Donnan potential to appear. The other option is the adsorption of hydroxyl anions present in water on polar monomer units of PAAm. It also can result in negative values of Donnan potential, which would increase with PAAm concentration. Both scenarios can work together as well.

For now, these are only hypothetical options that are to be verified in future studies. In the present study, we simply state that the PAAm chemical network provided measurable negative Donnan potential, which increased in absolute values with PAAm concentration in combined PAAm/CaAlg gels. In total, the largest negative value of Donnan potential (−62.6 mV) was achieved at the highest PAAm and CaAlg concentrations, and it was close to the physiological level of cell potential.

## 4. Conclusions

The physicochemical properties of hydrogels with interpenetrated physical and chemical networks were considered in relation to their prospective application as biomimetic materials in biomedicine and bioengineering. The study was focused on combined hydrogels with a chemical network of a synthetic polymer—polyacrylamide (PAAm) and a physical network of a natural polysaccharide—calcium alginate (CaAlg).

A thermodynamic study of a binary CaAlg/PAAm blend revealed that the enthalpy of interaction among PAAm and CaAlg was negative, which indicated the energetically favorable molecular contacts among PAAm and CaAlg monomer units. At the same time, the Gibbs energy of their blending was found to be positive, which meant the thermodynamic incompatibility of polymers. Both values were rather low as compared to thermal energy, but such a controversial combination of thermodynamic functions of mixing governed other physicochemical properties of combined PAAm/CaAlg gels. Thus, due to the energetic favorability of inter-chain contacts, the swelling ratio (water uptake) of hydrogels substantially diminished with the increase in the concentration of PAAm and CaAlg in the combined network. The effective mesh size of the chemical network of PAAm effectively decreased due to the extra cross-links provided by molecular contacts with CaAlg chains. At the same time, the general thermodynamic incompatibility of these two polymers resulted in almost complete re-swelling of combined gels after drying. Individual CaAlg gel, having been dried, recovered only about 5% of its initial water uptake. The combination of the physical network of CaAlg with the thermodynamically incompatible chemical network of PAAm made their dry blend thermodynamically unstable, which provided 80–100% recovery of swelling for IPN hydrogels.

Mechanical properties of combined hydrogels were also consistent with thermodynamic characteristics of molecular interactions among PAAm and CaAlg. At any composition, combined hydrogels showed the prevalence of the storage modulus over the loss modulus, which indicated the dominance of elastic deformation. While the values of storage modulus increased with the concentration of PAAm and CaAlg both, the impact of CaAlg content was much stronger than that of PAAm. In our opinion, it was a result of a combination of energetically favorable adhesion among polymers and their thermodynamic incompatibility. On the one hand, the adhesion of PAAm monomer units onto CaAlg chains favored the formation of additional cross-links in the double network. On the other hand, the thermodynamic incompatibility among CaAlg and PAAm led to extra stress applied to the conformation of rigid CaAlg subchains in the combined network. Both effects—the inter-chain adhesion and the conformational restrictions—favored the increase in the compression modulus. Such a combined effect might resemble the effect of fiber-loading into polymer composite on its mechanical properties.

The electrical testing of hydrogels showed the negative values of Donnan potential at any composition, including individual PAAm hydrogels. The negative sign of Donnan potential for CaAlg physical network was anticipated as it contained carboxylate residues which could provide the net negative charge on alginate macromolecules. The negative Donnan potential of the PAAm network cannot be readily explained as PAAm presumably is a neutral polymer. This result needs further experimental and theoretical clarification. In total, the negative Donnan potential increased in absolute values both with CaAlg and PAAm concentration in combined PAAm/CaAlg DN hydrogels. The largest negative value of Donnan potential (−62.6 mV) was achieved at the highest PAAm and CaAlg concentrations, and it was close to the physiological level of cell potential.

From the viewpoint of biomedical applications, IPN hydrogels with a combined polymer network have a number of advantages over systems based on a single chemical or physical network. In fact, the double polymeric network of the PAAm/CaAlg gel increases biocompatibility and reduces the biodegradation of the composite compared to alginate gel [23,24,25,26]. We have shown that such gels mimic well the resting potential in living cells, as well as the elastic properties of unexcited soft tissues where Young’s modulus is in the range of 15–50 kPa [47]. Qualitatively, the same results were obtained when combining polyacrylamide with other polysaccharides (xanthan or gellan) [48].

Notably, PAAm/CaAlg hydrogels can mimic the electromechanical and mechanoelectrical phenomena [49] in living excitable cells in a manner like that of other anionic polyelectrolyte gels [50,51]. In addition, the spatial structure of the PAAm/CaAlg hydrogel allows the introduction of magnetic particles into the composite [52], which potentially expands the functional properties of biomimetics with a double polymer network.

Thus, varying the percentage ratio of polyacrylamide and natural polysaccharide in the IPN hydrogel opens up opportunities for designing mimetics with useful properties that are closest to biological prototypes, e.g., to muscles.

## Figures and Tables

**Figure 1 biomimetics-08-00279-f001:**
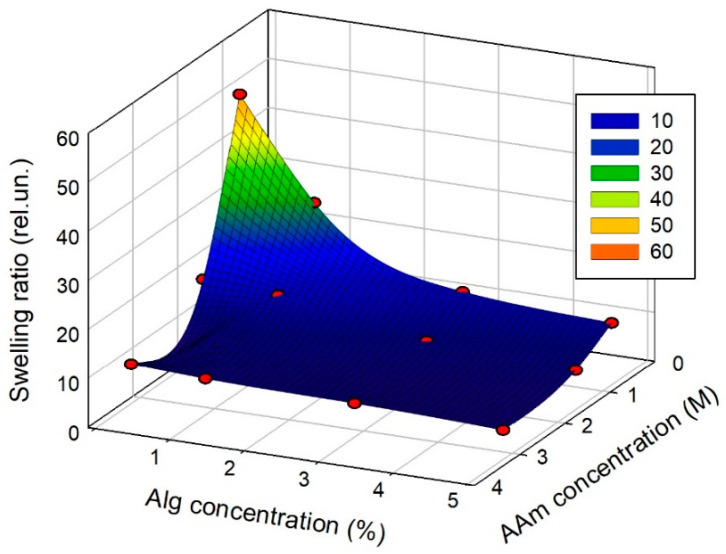
Dependence of the swelling ratio of PAAm/CaAlg hydrogels with interpenetrating chemical and physical networks on the concentration of alginate and AAm monomer in the reaction mixture for the synthesis.

**Figure 2 biomimetics-08-00279-f002:**
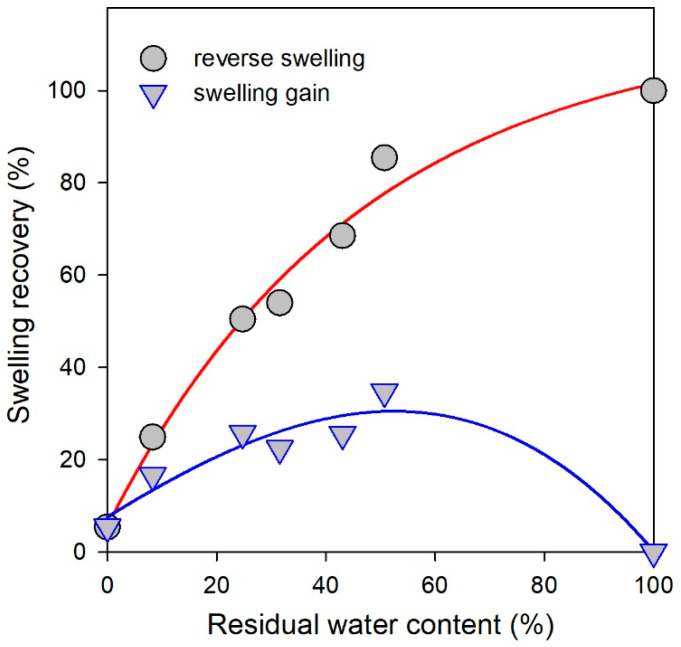
Dependence of the degree of swelling recovery on the residual water content left in CaAlg gel after its partial drying. Circles correspond to the integral recovery; triangles correspond to the additional gain in water content in re-swelling. Lines are for eye-guide only.

**Figure 3 biomimetics-08-00279-f003:**
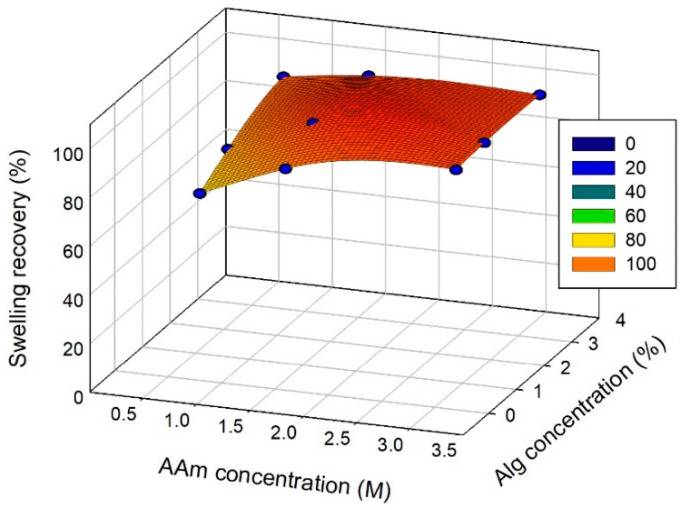
Recovery of the swelling ratio of PAAm/CaAlg gels with double network depending on their composition. AAm and alginate concentrations correspond to the composition of the reaction mixture for polymerization.

**Figure 4 biomimetics-08-00279-f004:**
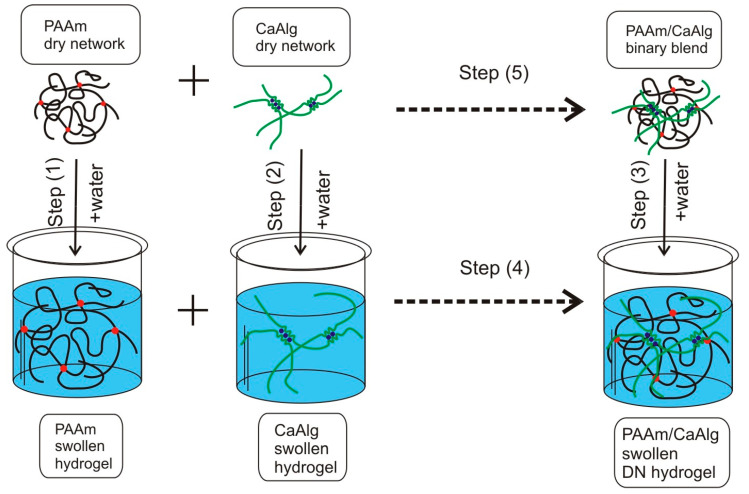
Thermodynamic cycle for the evaluation of thermodynamic functions of mixing for a binary polymer/polymer composition (blend).

**Figure 5 biomimetics-08-00279-f005:**
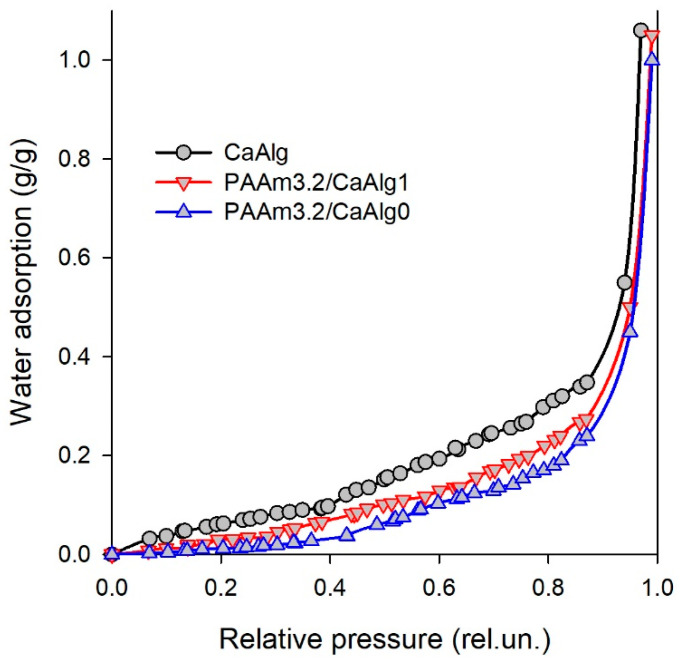
Isotherms of water vapor adsorption (25 °C) on dried gels. Water adsorption is expressed in g of water per 1 g of polymer.

**Figure 6 biomimetics-08-00279-f006:**
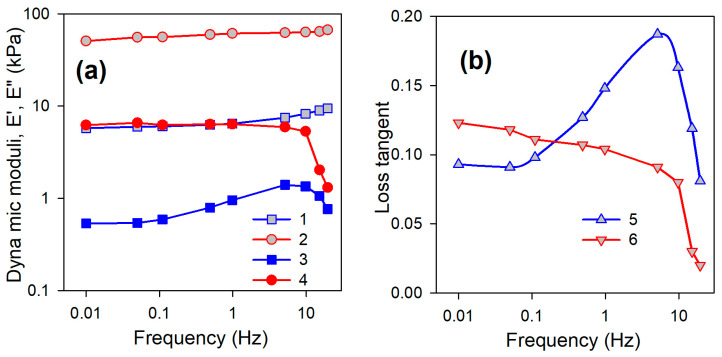
Frequency dependence of storage modulus, loss modulus (**a**), and loss tangent (**b**) for PAAm/CaAlg hydrogels. PAAm1.6/CaAlg0: 1—*E*′, 3—*E*″, 5—*tgδ*. PAAm1.6/CaAlg5: 2—*E*′, 4—*E*″, 6—*tgδ*.

**Figure 7 biomimetics-08-00279-f007:**
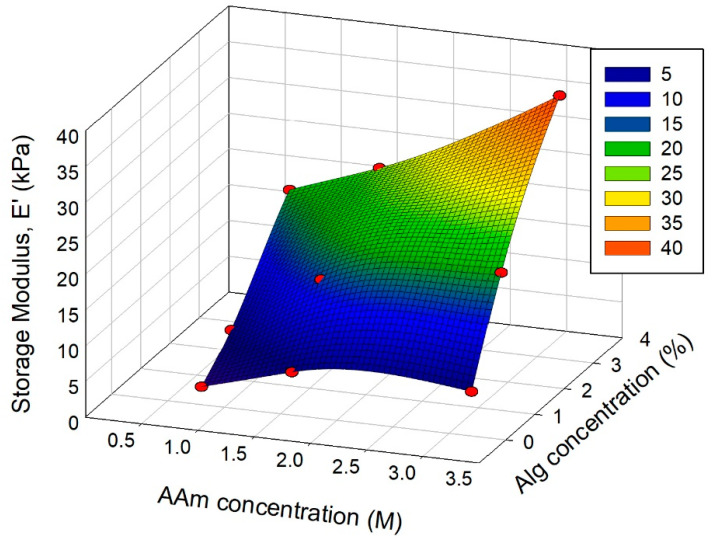
Storage modulus (*E*′) of PAAm/CaAlg gels with double network depending on their composition. Frequency 0.05 Hz, 25 °C. AAm and alginate concentrations correspond to the composition of the reaction mixture for polymerization. The relative error of modulus measurements was 15%.

**Figure 8 biomimetics-08-00279-f008:**
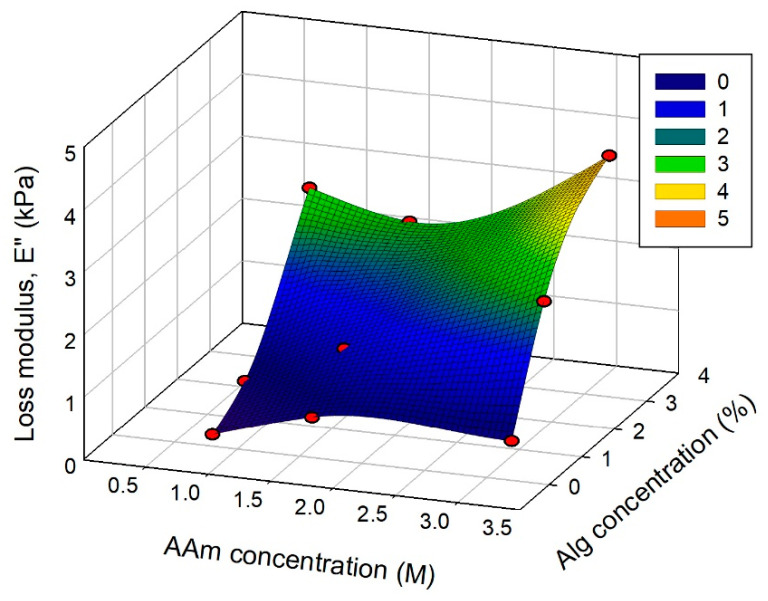
Loss modulus (*E*″) of PAAm/CaAlg gels with a double network depending on their composition. Frequency 0.05 Hz, 25 °C. AAm and alginate concentrations correspond to the composition of the reaction mixture for polymerization. The relative error of modulus measurements was 15%.

**Figure 9 biomimetics-08-00279-f009:**
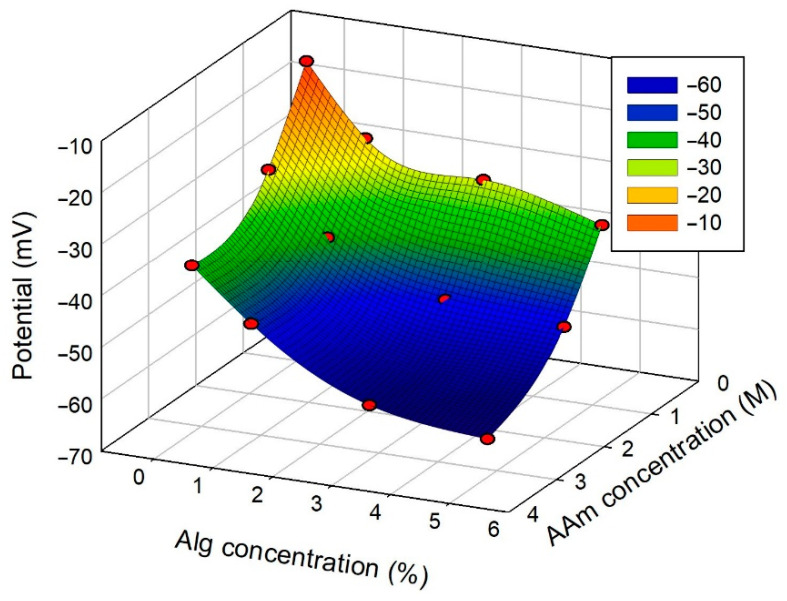
Donnan potential of PAAm/CaAlg gels with a double network depending on their composition. AAm and alginate concentrations correspond to the composition of the reaction mixture for polymerization. The relative error of potential measurements was 10%.

**Table 1 biomimetics-08-00279-t001:** Parameters of networking density for PAAm/CaAlg hydrogels with different compositions.

Gel	Swelling Ratio	PAAm Actual Concentration (%)	CaAlg Actual Concentration (%)	Volume Swelling Ratio Related to PAAm	*N_C_*
PAAm0.8/CaAlg0	48.0	2.0	0.00	61.0	1580
PAAm0.8/CaAlg1	28.1	2.9	0.51	40.6	730
PAAm0.8/CaAlg3	14.5	4.2	2.21	28.2	353
PAAm0.8/CaAlg5	12.7	3.9	3.39	30.2	406
PAAm1.6/CaAlg0	15.3	6.1	0.00	19.4	163
PAAm1.6/CaAlg1	14.3	6.0	0.52	19.1	158
PAAm1.6/CaAlg3	9.5	7.6	1.97	15.4	102
PAAm1.6/CaAlg5	8.1	7.7	3.32	14.9	96
PAAm3.2/CaAlg0	8.1	11.0	0.00	10.3	43
PAAm3.2/CaAlg1	7.4	11.4	0.49	9.8	39
PAAm3.2/CaAlg3	6.9	11.2	1.46	9.9	40
PAAm3.2/CaAlg5	6.0	11.7	2.54	9.3	35

**Table 2 biomimetics-08-00279-t002:** Thermodynamic functions for swelling of hydrogels in water and for PAAm/CaAlg compatibility, 25 °C.

Gel	Enthalpy of Swelling, 25 °C (J/g)	Enthalpy of Mixing, ∆*H_m_* (kJ/mol)	Chemical Potential of Polymer, ∆*μ_2_* (J/g)	Gibbs Energy of Mixing, ∆*G_m_* (kJ/mol)	Entropy of Mixing, *T*∆*S_m_* (kJ/mol)
CaAlg	−199.3 ± 3.2		−59.3 ± 1.2		
PAAm0.8/CaAlg0	−113.6 ± 3.2		−25.4 ± 0.8		
PAAm1.6/CaAlg0	−112.8 ± 3.1		−22.8 ± 0.8		
PAAm3.2/CaAlg0	−112.6 ± 1.8		−22.9 ± 0.6		
PAAm0.8/CaAlg1	−119.1 ± 2.9	−0.71 ± 0.09	−33.1 ± 0.9	0.12 ± 0.03	−0.83
PAAm1.6/CaAlg1	−112.5 ± 1.5	−0.62 ± 0.09	−28.1 ± 1.2	0.09 ± 0.03	−0.71
PAAm3.2/CaAlg1	−110.8 ± 1.2	−0.81 ± 0.09	−30.4 ± 0.8	0.39 ± 0.03	−1.20

## Data Availability

Not applicable.

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
