# Peer review of "Hydrogels Based on Polyacrylamide and Calcium Alginate: Thermodynamic Compatibility of Interpenetrating Networks, Mechanical, and Electrical Properties"

_biomimetics, 2023, doi:10.3390/biomimetics8030279_

Round 1
Reviewer 1 Report
The manuscript entitled “ Hydrogels Based on Polyacrylamide and Calcium Alginate: Thermodynamic Compatibility of Interpenetrating Networks, Mechanical, and Electrical Properties” refers to synthesis and properties of combined hydrogels based on calcium alginate and polyacrylamide with interpenetrated physical and chemical networks. Authors synthesized the series of hydrogels with varying amounts of alginate in the combined hydrogels.
Their water uptake was investigated within the equilibrium swelling and re-swelling. Moreover, the compatibility of alginate and polyacrylamide was studied by the thermodynamic method and also their mechanical and electrical properties.
The studies are valuable in the point of view the hydrogels prospective application as biomimetic materials in biomedicine and bioengineering.
The manuscript is basically well written and suitable for readers of Biomimetics.
I have a few minor comments.
1. The purity of reagents was omitted in the section 2.
2. Did the authors make some statistical analysis of the measured properties?
I think that the manuscript should be published after minor correction.
Author Response
We appreciate the comments made by Reviewers. They were profound, friendly, and aimed to improve the presentation of our results. The revised version of the manuscript was edited according to all comments made. Our corrections to the text are marked in yellow. We have also corrected Figure 4. Below are our responses to the comments.
Reviewer 1:
The manuscript entitled “ Hydrogels Based on Polyacrylamide and Calcium Alginate: Thermodynamic Compatibility of Interpenetrating Networks, Mechanical, and Electrical Properties” refers to synthesis and properties of combined hydrogels based on calcium alginate and polyacrylamide with interpenetrated physical and chemical networks. Authors synthesized the series of hydrogels with varying amounts of alginate in the combined hydrogels.
Their water uptake was investigated within the equilibrium swelling and re-swelling. Moreover, the compatibility of alginate and polyacrylamide was studied by the thermodynamic method and also their mechanical and electrical properties.
The studies are valuable in the point of view the hydrogels prospective application as biomimetic materials in biomedicine and bioengineering.
The manuscript is basically well written and suitable for readers of Biomimetics.
I have a few minor comments.
- The purity of reagents was omitted in the section 2.
Responce
Corrected
- Did the authors make some statistical analysis of the measured properties?
Responce
Statistical analysis was done for measured thermodynamic properties as it was given in table 2. Statistics was as well provided for dynamic moduli (Figures 6 - 8) and Donnan potential (Figure 9). The error bars were not shown in the plots. We have added the average estimation of relative error to the captions of these figures.
I think that the manuscript should be published after minor correction.
Reviewer 2 Report
In the manuscript by Safronov et al., the authors studied a well-known double network composed of a physically crosslinked calcium alginate network and a covalently crosslinked polyacrylamide network, examining its thermodynamic behaviors, mechanical, and electrical properties. Utilizing Flory-Rehner theory and calorimetric measurements, the authors attempted to understand the swelling behaviors and thermodynamic parameters of the interpenetrated network. Further, they studied the mechanical and electrical properties of the double network to broaden its potential application as cell culturing scaffolds and biomimicry actuators. Upon first reading the manuscript, this work struck me as fundamentally interesting given the fact that thermodynamic studies on double networks are both rare and challenging. However, after thoroughly examining the manuscript, this reviewer must politely point out that the thermodynamic studies performed appear somewhat superficial and, at times, potentially misleading. Some major concerns include:
(1) Flory-Rehner theory, established based on chemically crosslinked networks like vulcanized rubber (J. Chem. Phys. 1943, 11, 521), is risky to apply to a physically crosslinked network. The determination of Flory-Huggins interaction parameters is difficult when polymer-polymer interaction is significant and even dominant. In a physically crosslinked network, the degree of polymerization also needs a precise definition to accurately reflect the network connectivity.
(2) Things become even more unpredictable when a more complex system, like the double network mentioned in this work, is introduced. In a double network, multiple interactions coexist, including polymer1-polymer1 interaction, polymer2-polymer2 interaction, polymer1-polymer2 interaction, polymer1-solvent interaction, and polymer2-solvent interaction. The existence of multiple types of crosslinking suggests that the simplifications made in the manuscript could be an oversight.
(3) The descriptions and interpretations of thermodynamic characterizations from calorimetry are questionable. The chart in Figure 4 seems reasonable, but Step (5) involves the mixing of the dry PAAm network and dry Ca Alginate network, which does not correspond to the mixing of two networks in the presence of water. Additionally, the mixing of linear polymers depicted in Step (4) does not necessarily represent the mixing of two networks. The authors may find this paper of use for their revisions: J. Phys. Chem. B 2008, 112, 3903.
Overall, while the experimental part of the manuscript is intriguing, the data analysis and thermodynamic arguments present major issues. This reviewer recommends that the authors remove those questionable statements relating to thermodynamics and instead place a greater focus on experimental observations such as swelling behavior and enthalpy of swelling. Upon addressing these issues, I might recommend the publication of this manuscript.
Here are a few additional technical comments:
(1) In the Flory-Rehner equation, the authors used mass fraction, yet the equation was derived based on polymer volume fraction. Is this an oversight or did the authors measure the polymer density before performing the calculation?
(2) The preparation of a pure calcium alginate network was not described in the manuscript. This reviewer suggests including these experimental details.
(3) The storage modulus and the loss modulus were denoted by two different letters G'/G'' and E'/E''. They should be represented as E' and E'' based on the experimental technique the authors used.
(4) The representation of calcium ion should be Ca2+, not Ca++, and potassium ion should be K+.
(5) While the language is comprehensible, it could be more reader-friendly. Some sentences conveying simple concepts are written in a somewhat obfuscated way. For example, the sentence “Meanwhile, the increase of AAm monomer concentration in the synthesis led to the increase in the probability of the geometric entanglement of growing PAAm chains which was equivalent to the formation of spatial cross-links in addition to the chemical cross-links. As a result, the actual networking density deviated from the level pre-set in the synthesis.”, could be simplified to: “Polymer entanglements are more likely to happen at a higher AAm monomer concentration, increasing the actual crosslinking density by introducing spatial cross-links in addition to the chemical cross-links.”
Please see the comments above for details. Thank you.
Author Response
We appreciate the comments made by Reviewers. They were profound, friendly, and aimed to improve the presentation of our results. The revised version of the manuscript was edited according to all comments made. Our corrections to the text are marked in yellow. We have also corrected Figure 4. Below are our responses to the comments.
In the manuscript by Safronov et al., the authors studied a well-known double network composed of a physically crosslinked calcium alginate network and a covalently crosslinked polyacrylamide network, examining its thermodynamic behaviors, mechanical, and electrical properties. Utilizing Flory-Rehner theory and calorimetric measurements, the authors attempted to understand the swelling behaviors and thermodynamic parameters of the interpenetrated network. Further, they studied the mechanical and electrical properties of the double network to broaden its potential application as cell culturing scaffolds and biomimicry actuators. Upon first reading the manuscript, this work struck me as fundamentally interesting given the fact that thermodynamic studies on double networks are both rare and challenging. However, after thoroughly examining the manuscript, this reviewer must politely point out that the thermodynamic studies performed appear somewhat superficial and, at times, potentially misleading. Some major concerns include:
(1) Flory-Rehner theory, established based on chemically crosslinked networks like vulcanized rubber (J. Chem. Phys. 1943, 11, 521), is risky to apply to a physically crosslinked network. The determination of Flory-Huggins interaction parameters is difficult when polymer-polymer interaction is significant and even dominant. In a physically crosslinked network, the degree of polymerization also needs a precise definition to accurately reflect the network connectivity.
Responce 1.
Yes, we agree that Flory-Rehner theory as it had been introduced had addressed the chemical networks of flexible polymers and it could not be readily applied to physical networks. However, the history of Flory-Huggins theory showed that it had not once been extended far beyond the limits of its initial scope. In this respect we believe that it is worthwhile to test its applicability even at more complex systems if it would give a reasonable outcome. Actually, in the paper we had implicitly used the model of a flexible chemical network of PAAm filled with polymeric filaments (Alg) which interacted with water only at their surface. Thus, the whole water uptake was related to the swelling of PAAm network. It was a simplification which was, obviously, not presented in the text in a proper way. The aim of this assumption was to give a feasible qualitative explanation for the trends in swelling ratio (Figure 1) based on the classical theory. We have edited the text to stress out the problems and the simplifications of the presented approach.
(2) Things become even more unpredictable when a more complex system, like the double network mentioned in this work, is introduced. In a double network, multiple interactions coexist, including polymer1-polymer1 interaction, polymer2-polymer2 interaction, polymer1-polymer2 interaction, polymer1-solvent interaction, and polymer2-solvent interaction. The existence of multiple types of crosslinking suggests that the simplifications made in the manuscript could be an oversight.
Responce 2.
It is correct. Certainly, we understand that the presented evaluation is by a necessity simplified. A tri-component system includes all these types of interaction. Some words about that we had given in the initial version concerning the evaluation of thermodynamic compatibility which included the switch from tri-component DN hydrogel to a binary polymer blend. But we agree that the same consideration would also be provided concerning the analysis of swelling ratios based on Flory-Rehner approach. Apart from the problems which were addressed in Responce 1 multiple interactions affect the apparent value of Flory-Huggins interaction parameter which was used in Equation (2). Therefore, the calculated Nc values should not be taken as an accurate but rather effective estimation. Meanwhile, the trends in Nc do not depend on the exact value of Flory-Huggins parameter as it is a constant in Equation (2) and the only variable is the swelling ratio. We have edited the text concerning evaluation of Nc to emphasize the uncertainty of binary Flory-Huggins parameter in DN hydrogels and the effective character of Nc values.
(3) The descriptions and interpretations of thermodynamic characterizations from calorimetry are questionable. The chart in Figure 4 seems reasonable, but Step (5) involves the mixing of the dry PAAm network and dry Ca Alginate network, which does not correspond to the mixing of two networks in the presence of water. Additionally, the mixing of linear polymers depicted in Step (4) does not necessarily represent the mixing of two networks. The authors may find this paper of use for their revisions: J. Phys. Chem. B 2008, 112, 3903.
Response 3.
It is correct that the chart in Figure 4 does not represent thermodynamic functions of mixing for the swollen networks but the functions for polymeric blend of dry networks. Although it was mentioned at the beginning of section 3.2, obviously it was not stated clearly enough and left space for different assumption. We have emphasized relation to the polymer binary blend in the title of the section and have extended discussion on that in the text of the section. Figure 4 is also edited and schematic images of networks and gels are provided for guidance. We agree, that mixing of linear polymers does not necessarily represent mixing of two swollen networks. However, in regard to the cycle presented in Figure 4, it is the only feasible approximation as the process of mixing of two networks each being already swollen in water is physically impossible. As a matter of fact, the mixing of dry networks is also physically impossible and cannot be performed without disruption of these networks. In this respect, the target step (5) in the cycle is a virtual process, which in fact stands for the interaction of Alg and PAAm sub-chains inside the double network. We have added discussion on that to the text of the section. We appreciate the recommended reference. We have included it to the list with appropriate references in the text.
Overall, while the experimental part of the manuscript is intriguing, the data analysis and thermodynamic arguments present major issues. This reviewer recommends that the authors remove those questionable statements relating to thermodynamics and instead place a greater focus on experimental observations such as swelling behavior and enthalpy of swelling. Upon addressing these issues, I might recommend the publication of this manuscript.
Responce 4
Thermodynamic approach aimed on the evaluation of thermodynamic functions of mixing in multi-component polymeric systems had been introduced decades ago in works by prof. Anna Tager in former USSR starting from early 1970-ies. It included consideration of polymer blends and filled polymers using appropriate thermodynamic cycles. Those reports, however, might not be well presented at international level at that time. Nevertheless, there is a solid experimental background for such an approach. Most recent international publications on synthetic/polysaccharide polymer blends are: J. Polym. Sci. B. Polym. Phys. 2007, V.45, N18, P. 2603-2613 (Copolyamide/Chitosan blend); Polymer Science, Series A, 2022, Vol. 64, No. 1, pp. 53–62 (Agarose/PAAm blend). The former, besides, includes references to the earlier works. In this respect experimental thermodynamic measurements since prof. Tager works have made themselves sort of a scientific brand for our group in Ural Federal University. We believe it a useful and versatile tool in polymer physics of solutions, blends, and filled compositions. We have added brief general introduction of this approach to the text with these two references.
In the present work we have tried to extend the approach to polymeric gels with double network. There are definite dissimilarities with linear polymers but there are also common features. We have edited the text to point out problems, assumptions and simplifications in this respect. To compare thermodynamic compatibility of the networks with that of linear polymers we are planning a research on linear NaAlg / linear PAAm blend in nearby future.
Here are a few additional technical comments:
(1) In the Flory-Rehner equation, the authors used mass fraction, yet the equation was derived based on polymer volume fraction. Is this an oversight or did the authors measure the polymer density before performing the calculation?
Response
We agree. Correct use of Flory-Rehner equation implies volume swelling ratios. In the initial version we have neglected the difference between weight fraction and volume fraction. In the revised version it was corrected. Necessary explanations in the text are given, the values in Table 1 are recalculated.
(2) The preparation of a pure calcium alginate network was not described in the manuscript. This reviewer suggests including these experimental details.
Response
Included.
(3) The storage modulus and the loss modulus were denoted by two different letters G'/G'' and E'/E''. They should be represented as E' and E'' based on the experimental technique the authors used.
Response
Corrected
(4) The representation of calcium ion should be Ca2+, not Ca++, and potassium ion should be K+.
Responce
Corrected
(5) While the language is comprehensible, it could be more reader-friendly. Some sentences conveying simple concepts are written in a somewhat obfuscated way. For example, the sentence “Meanwhile, the increase of AAm monomer concentration in the synthesis led to the increase in the probability of the geometric entanglement of growing PAAm chains which was equivalent to the formation of spatial cross-links in addition to the chemical cross-links. As a result, the actual networking density deviated from the level pre-set in the synthesis.”, could be simplified to: “Polymer entanglements are more likely to happen at a higher AAm monomer concentration, increasing the actual crosslinking density by introducing spatial cross-links in addition to the chemical cross-links.”
Responce
Correction is appreciated. Text was additionally checked and edited.
Round 2
Reviewer 2 Report
The authors have addressed this reviewer's comments. This reviewer would like to recommend the publication of this manuscript.